# Oligotrophic Gene Expression in *Rhodococcus erythropolis* N9T-4 under Various Nutrient Conditions

**DOI:** 10.3390/microorganisms10091725

**Published:** 2022-08-27

**Authors:** Yuri Ikeda, Mana Kishimoto, Masaki Shintani, Nobuyuki Yoshida

**Affiliations:** 1Applied Chemistry and Biochemical Engineering Course, Department of Engineering, Graduate School of Integrated Science and Technology, Shizuoka University, Hamamatsu 432-8561, Japan; 2Research Institute of Green Science and Technology, Shizuoka University, 836 Ohya, Suruga-ku, Shizuoka 422-8529, Japan

**Keywords:** oligotrophs, methanol dehydrogenase, alcohol dehydrogenase, gene expression

## Abstract

*Rhodococcus erythropolis* N9T-4 is a super oligotroph that grows on an inorganic basal medium without any additional carbon and nitrogen sources and requires CO_2_ for its oligotrophic growth. Previously, we found that two genes, *aldA* and *mnoA,* encoding NAD-dependent aliphatic aldehyde dehydrogenase and *N*,*N*′-dimethyl-4-nitrosoaniline-dependent methanol dehydrogenase, respectively, were highly upregulated under oligotrophic conditions. In this study, we constructed reporter plasmids containing an enhanced green fluorescent protein gene under *aldA* or *mnoA* promoters (pAldA and pMnoA, respectively). Fluorescence analysis of N9T-4 cells with reporter plasmids revealed that tryptone and yeast extract strongly repressed the expression of oligotrophy-connected genes, whereas the effect of casamino acids was moderate. Furthermore, remarkably high expression of *aldA* and *mnoA* was observed when the reporter strains were grown in media containing primary alcohols, particularly ethanol. Malic acid repressed ethanol-induced gene expression, suggesting that C2 metabolism is involved in the oligotrophic growth of N9T-4. The regulation of oligotrophic gene expression elucidated in this study could provide appropriate conditions for the production of useful compounds in an oligotrophic microbial process.

## 1. Introduction

We designated microorganisms that can grow on an inorganic minimum medium (BM medium) without the addition of any carbon or energy sources as “super oligotrophs,” and have tried to isolate such bacteria from various natural environments [1]. Intriguingly, super oligotrophs can be easily isolated from different environments, suggesting that they play a vital role in the nutrient cycles of terrestrial ecosystems because most natural environments must be oligotrophic. However, there may be no sense in arguing whether the nutrient levels for microbial growth are high or low in natural environments, as low nutrient concentrations would be sufficient for the growth of oligotrophs. Optimum nutrient conditions in flasks are not necessarily appropriate for bacterial growth in the original natural environment. We should distinguish between laboratory “nutrients” and natural environment “nutrients”.

One of the aims of collecting various types of super oligotrophs involves their industrial applications. Super oligotrophs can grow on nutrient-poor media, leading to lower cultivation costs for the industrial production of useful compounds. Furthermore, oligotrophic features may be appropriate for in situ bioremediation because additional nutrients for bacterial growth may not be needed at the contaminant sites. We can use super oligotrophs for their intrinsic metabolism or as hosts expressing useful heterologous genes, providing low-cost and low-energy bioprocesses in the future. An understanding of the nutrient conditions is needed to determine those specialized for the industrial production of useful compounds.

*Rhodococcus erythropolis* N9T-4, isolated from the stocked crude oil in an oil stockpile in Japan, showed the best oligotrophy among the super oligotrophs that we isolated so far and grew well on a solid-state BM medium without any additional carbon and nitrogen sources [2,3,4]. This bacterium could not grow under CO_2_-limiting conditions, where a CO_2_ absorbent was added to a plastic bag, and the growth was recovered when CO_2_ gas was injected into the plastic bag, or carbon sources, such as sodium carbonate or *n*-tetradecane, were added to the medium. These results suggest that this bacterium uses CO_2_ as a carbon source for oligotrophic growth. However, the CO_2_ fixation system has not been proven, and currently, its CO_2_ requirement is not involved in the main carbon metabolism of N9T-4, while CO_2_ incorporation has been confirmed in the cell components [5]. We suggest that alternative-specific carbon metabolism is involved in the oligotrophic growth of N9T-4. Our previous proteomic analysis showed that several proteins, such as NAD-dependent aliphatic aldehyde dehydrogenase (NAD-ALDH), *N*,*N*′-dimethyl-4-nitrosoaniline-dependent methanol dehydrogenase (NDMA-MDH), and chaperone-related proteins, were highly upregulated under oligotrophic conditions [3]. Furthermore, DNA microarray analysis revealed that two genes, *aldA* and *mnoA*, encoding NAD-ALDH and NDMA-MDH, respectively, were strongly expressed under oligotrophic conditions [6]. All the super oligotrophs that we isolated previously belonged to the genera *Rhodococcus* and *Streptomyces* [1,5], and Nagata found that a transposon-induced mutant of *Sphingobium japonicum* UT26 showed oligotrophic features, and a Zn-dependent alcohol dehydrogenase was essential for oligotrophic growth [7].

There are several questions regarding the oligotrophy of N9T-4. What are the carbon and energy sources for the oligotrophic growth of N9T-4? Why are *aldA* and *mnoA* remarkably upregulated under oligotrophic conditions? To answer these questions, we elucidated the oligotrophic conditions for N9T-4 using *aldA* and *mnoA* as oligotrophic indicators. Previously, we identified promoter regions for *aldA* and *mnoA* in the N9T-4 genome [8]. We constructed plasmids containing a structural gene encoding the enhanced green fluorescent protein (EGFP) under the control of pAld or pMnoA. N9T-4 cells with reporter plasmids were cultivated under various nutrient conditions, and the fluorescence of the cells was measured.

## 2. Materials and Methods

### 2.1. Materials

The materials used for media preparation were tryptone (Nacalai Tesque, Kyoto, Japan), yeast extract (Oriental Yeast, Tokyo, Japan), peptone (Hi Polypepton, Nihon Pharmaceutical, Tokyo, Japan), and casamino acids (Hi Casamino Acids Daigo, Nihon Pharmaceutical, Tokyo, Japan).

### 2.2. Bacterial Strain and Cultivation Media

The *R. erythropolis* N9T-4 Δ*ligD* strain, showing lower non-homologous recombination of exogenous genes [9], was used throughout this study as a host strain of each reporter plasmid and was cultivated under oligotrophic or eutrophic conditions to measure its fluorescence. A BM medium consisting of 1 g/L NaNO_3_, 1 g/L K_2_HPO_4_, 1 g/L KH_2_PO_4_, 0.5 g/L MgSO_4_•H_2_O, 0.1 g/L CaCl_2_•2H_2_O, and 1 µg /L thiamin-HCl (pH 7.0) [3] was used for oligotrophic cultivation, and carbon and nitrogen sources were added to the BM medium as appropriate. The Luria–Bertani (LB) medium and 2 x LB medium, in which concentrations of tryptone and yeast extract were twice higher than those in the LB medium, were used for plasmid construction using *Escherichia coli* DH5α and heterotrophic cultures of N9T-4, respectively.

### 2.3. Construction of Reporter Plasmids

The DNA fragment containing the EGFP gene was synthesized artificially (Eurofins Genomics, Tokyo, Japan) and amplified by PCR using the primer set EGFP_fwd (5′-gga gct tgc aat ggt gag caa ggg cga g-3′) and EGFP_rev (5′-cac ggg tgc cgg tgg gtc gat tac ttg tag agc tcg tcc atg-3′) for the 5′- and 3′-regions, respectively. A 200-bp upstream region of *aldA* was also amplified by PCR using the primer set aldAup200_fwd (5′-cgc ggc gag tcc cca tgc ttc tgt att cgc agg tcc ag-3′) and aldAup200_rev (5′-tgc tca cca ttg caa gct cct atg taa ac-3′) for the 5′- and 3′-regions of *aldA*, respectively. The two PCR fragments were introduced to the BsrGI and SpeI sites in the plasmid pNit-RK2, constructed by replacing *chlR* (chloramphenicol resistance gene) with *kanR* (kanamycin resistance gene) in pNit-RC2 [10], using a NEBuilder HiFi DNA assembly (New England Biolabs Japan, Tokyo) to construct pAldA-EGFP-RK2. The reporter plasmid with a 280-bp *mnoA* promoter region, was also constructed in the same manner using the primer set mnoAup280_fwd (5′-cgc ggc gag tcc cca tgc ttc tag aac gtg ttc tag ttc-3′) and mnoAup280_rev (5′-cag tga tgg tga tgg tga tgg aag tac gac ttg ttg atc-3′) for the 5′- and 3′-regions of *mnoA*, respectively, to construct pMnoA-EGFP-RK2. Δ*ligD* competent cells were transformed with these plasmids by electroporation as described previously [9].

### 2.4. Fluorescence Intensity Measurement of Cells

The preculture was performed overnight at 30 °C in the LB broth supplemented with 50 μg/mL kanamycin. The cells were washed three times with 0.85% KCl and inoculated into the appropriate media containing 50 μg/mL kanamycin at an OD_660_ of 0.01. After cultivation at 30 °C for 12–48 h, cell fluorescence was measured using a fluorometer (Quantus Fluorometer, Promega K. K., Tokyo, Japan). The cells grown in the medium with autofluorescence were washed three times with 0.85% KCl before fluorescence measurement.

## 3. Results and Discussion

### 3.1. Evaluation of Reporter Plasmids in N9T-4 Cells

As described above, two oligotrophic genes were remarkably upregulated under oligotrophic conditions; we identified their promoter regions within the 200- and 280-bp upstream regions of *aldA* and *mnoA*, respectively [8]. In this study, we examined the expression of these oligotrophic genes using the EGFP reporter system in liquid cultures under various nutrient conditions. The growth of N9T-4 in a BM-liquid medium without any additional carbon source was quite low; however, this bacterium could grow well on a BM solid-state medium solidified with agar or silica gel [3]. We found that submerged cultivation of N9T-4 using a polyurethane foam sponge enhanced oligotrophic growth by ten times or more in a BM-liquid medium [11]. To evaluate the feasibility of the reporter system constructed in this study, N9T-4 cells harboring the plasmids pAldA-EGFP-RK2, pMnoA-EGFP-RK2, and pNit-RK2, respectively, were cultivated in a BM-submerged culture medium (12 mL of BM medium with 6 cm × 6 cm × 2.5 cm rectangular parallelepiped sponge), and the fluorescence intensities of the cells collected by squeezing the sponge were measured. The cells harboring pAldA-EGFP-RK2 or pMnoA-EGFP-RK2 showed approximately 30 times the intensity of fluorescence compared with those with pNit-RK2. Oligotrophic expression of EGFP under *aldA* and *mnoA* promoters was also confirmed when N9T-4 cells harboring the plasmids described above were cultivated on BM plates (Figure 1).

### 3.2. Gene Expression under Oligotrophic and Eutrophic Conditions

Because our previous transcriptional analysis by real-time PCR revealed that glucose did not affect the expression of *aldA* and *mnoA* in an induction experiment using a BM medium [6], we first compared *aldA* and *mnoA* expression in a BM medium containing glucose and a 2 × LB medium. As shown in Figure 2, the expression of these genes, in particular *mnoA*, was strongly repressed in the LB medium. Tryptone or yeast extract, the constituents of the LB medium, also repressed the two genes, whereas casamino acids seemed to enhance their expression. Tryptone and casamino acids are hydrolysates of casein produced by trypsin digestion and acid hydrolysis, respectively, and l-tryptophan in casamino acids must be decomposed by acid treatment during preparation. However, the effect of l-tryptophan on the expression of these two genes was relatively low, suggesting that peptides in tryptone and yeast extract affected *aldA* and *mnoA* expression.

From another viewpoint, the fluorescence intensities tended to be low when the reporter strains showed good growth, except for casamino acids. Cell density may be a signal for eutrophication and represses oligotrophic metabolism in N9T-4. To explore this possibility, the reporter strains were cultivated for 12 h in a BM medium containing yeast extract and showed the same levels of growth as those cultivated in a BM medium without a carbon source (each OD_660_ was approximately 0.1). As a result, the fluorescence intensities of the two reporter strains were almost the same as those cultivated for 24 h in the same medium, indicating that oligotrophic gene expression was not influenced by bacterial cell density but by some compounds in the cultivation medium.

### 3.3. Ethanol Induced Remarkably Oligotrophic Genes

NDMA-MDH catalyzes the oxidation of methanol to formaldehyde in Gram-positive methylotrophic bacteria [12,13]. We elucidated that NAD-ADH, encoded by *aldA,* catalyzed NAD-dependent dehydrogenation of various aldehydes, and the highest activity was observed against formaldehyde [3]. Based on these results, we first hypothesized that methanol metabolism is involved in the oligotrophic growth of N9T-4, and characterization of NDMA-MDH in N9T-4 is now in progress. Since NDMA-MDH in Gram-positive bacteria is active against methanol, ethanol, *n*-propanol, and *n*-butanol [13], the effect of these primary alcohols on oligotrophic gene expression was examined. As shown in Figure 3, a remarkable expression of *aldA* and *mnoA* was observed when grown in a BM medium containing ethanol. N9T-4 did not grow in the BM medium containing methanol, and the fluorescence could not be measured, suggesting that N9T-4 does not have a formaldehyde fixation pathway, which is a requisite for methylotrophy. Using ethanol-containing BM medium, oligotrophic gene repression under eutrophic conditions could be analyzed clearly more (Figure 4). Tryptone and yeast extract strongly repressed gene expression even when grown in a BM medium containing ethanol. Glucose and casamino acids showed moderate repression of oligotrophic genes, suggesting again that the other compounds in tryptone and yeast extract, such as peptides, except free amino acids, are involved in switching from oligotrophic to eutrophic metabolism.

Another intriguing result was that malic acid strongly repressed both the oligotrophic genes (Figure 4). Previously, we found that *aceA* and *aceB,* encoding isocitrate lyase and malate synthase, respectively, are essential for the oligotrophic growth of N9T-4, indicating that the glyoxylate shunt is the key to oligotrophy [9]. The accumulation of malate in the cells may imply stagnation of oligotrophic metabolism and cause, in particular, the remarkable repression of *mnoA*.

## 4. Conclusions

Although carbon metabolism of N9T-4 under oligotrophic conditions is still unclear, in this study, fluorescent reporter gene assays revealed that the oligotrophic genes, *aldA* and *mnoA,* were strongly expressed by primary alcohols, in particular ethanol. This suggests that C2 metabolism is involved in the oligotrophy of N9T-4 and is consistent with the fact that the glyoxylate shunt is essential for oligotrophic growth. Furthermore, it has been suggested that strong natural inhibitors exist in tryptone and yeast extract. The plasmids constructed in this study could also be used for the expression of genes involved in the production of useful compounds in an oligotrophic microbial process, and casamino acids in the cultivation medium would be effective for the expression, as they enhanced the growth of N9T-4 and did not affect oligotrophic gene expression in this study.

## Figures and Tables

**Figure 1 microorganisms-10-01725-f001:**
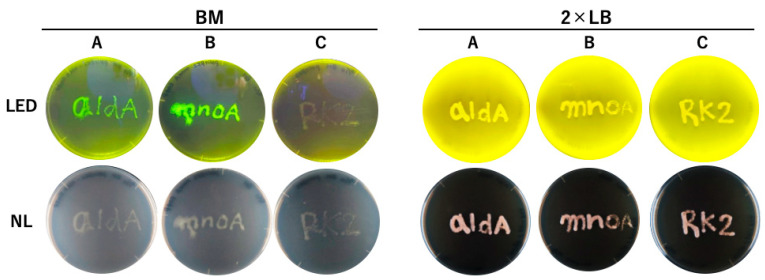
Construction of the reporter strains to examine *aldA* and *mnoA* expression. Several characters were drawn on a BM medium by streaking of N9T-4 Δ*ligD* cells having pAldA-EGFP (**A**), pMnoA-EGFP (**B**), or pNit-RK2 (**C**) and cultivated at 30 °C for 5 days. LED irradiation onto plates excited the fluorescence of EGFP.

**Figure 2 microorganisms-10-01725-f002:**
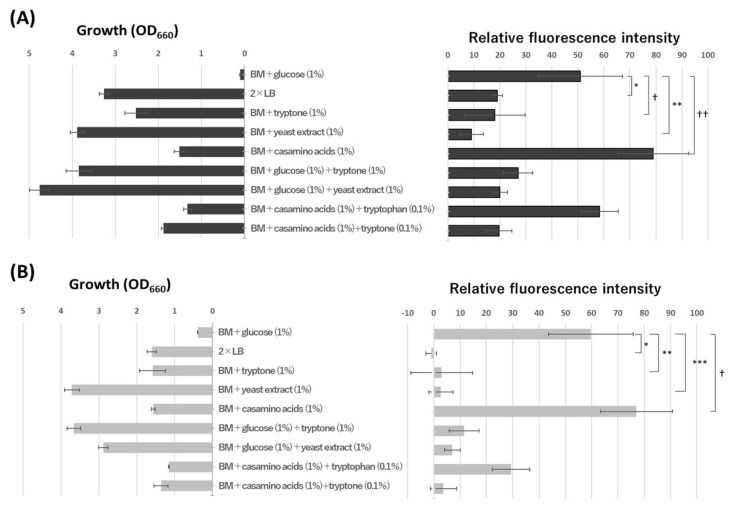
Oligotrophic gene expression under oligotrophic and eutrophic conditions. Washed cells of each reporter strain were inoculated into various media indicated in the figure at OD_660_ = 0.01. After cultivation for 24 h at 30 °C, the fluorescence intensity of the cells was measured as described in the text. The value of each strain having pAldA-EGFP (**A**) or pMnoA-EGFP (**B**) as a reporter plasmid was obtained by subtracting that of the control strain having pNit-RK2 grown under the same conditions. The values are the means of triplicate experiments, and the error bars show the standard deviations. Statistical differences were ascertained by Student’s *t*-test, *, **, and *** represent *p* < 0.05; † and †† represent *p* > 0.09.

**Figure 3 microorganisms-10-01725-f003:**
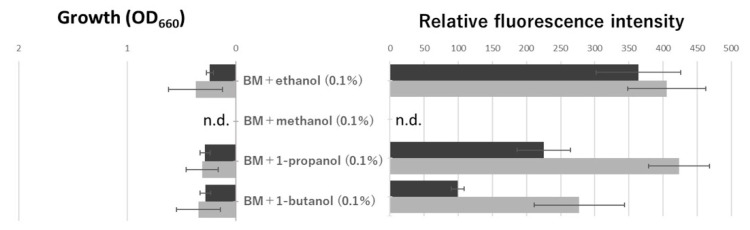
Oligotrophic gene expression in the medium containing various primary alcohols. The fluorescence intensity of the cells was measured as described in the legend in Figure 2. The cultivation time was 48 h.

**Figure 4 microorganisms-10-01725-f004:**
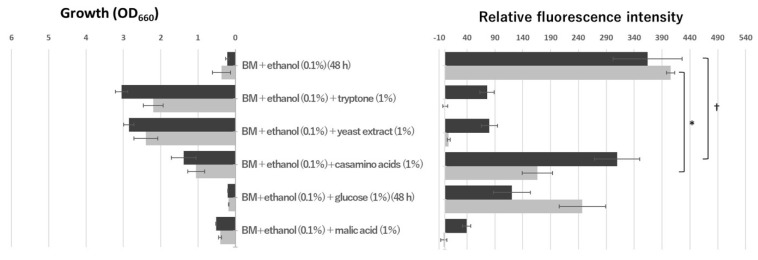
Effect of nutrients on the oligotrophic gene expression under ethanol-induced conditions. The fluorescence intensity of the cells was measured as described in the legend in Figure 2. The cultivation time was 24 h unless otherwise stated. Statistical differences were ascertained by Student’s *t*-test, * represents *p* < 0.05; † represents *p* > 0.09.

## Data Availability

The data presented in this study are available from the corresponding author.

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
