# Peer review of "Oligotrophic Gene Expression in Rhodococcus erythropolis N9T-4 under Various Nutrient Conditions"

_microorganisms, 2022, doi:10.3390/microorganisms10091725_

Round 1

Reviewer 1 Report

The topic is interesting, but the following issues need to be addressed to improve the article.

Comments

1.       Line 86, describe the composition of the BM medium in the text, as it is very important for repeating the experiment.

2.       Line 124-125,check the dimension unit of the sponge. cm or mm?

3.       Line 134-135, describe the concentration of glucose in the BM medium.

4.       Line 193-194, Figure 2, error bars for the value of OD660?

5.       Line 204-205 describe the concentration of ethanol, methanol, 1-propanol, 1-butanol.

6.       Line 208-209, describe the concentration of the substrate added in the BM.

Author Response

To reviewer 1

Thank you very much for your useful comments concerning our manuscript. We have carefully revised the manuscript based on your suggestions.

Below, please find our responses to your specific comments.

  1. Line 86, describe the composition of the BM medium in the text, as it is very important for repeating the experiment.

Response: Thank you for your appropriate suggestion The composition of BM medium has been described as follows in the revised text (line 87-88).

  1. Line 124-125, check the dimension unit of the sponge. cm or mm?

Response: Actually, we used a piece of 6 cm × 6 cm × 2.5 cm rectangular parallelepiped sponge on the bottom of a beaker, which was dipped with BM medium.

  1. Line 134-135, describe the concentration of glucose in the BM medium.

Response: The concentration of glucose in BM medium was 1% (Figure 2). We have also added the concentrations of the other components used for BM medium in the figures. Thank you for your valuable suggestion.

  1. Line 193-194, Figure 2, error bars for the value of OD660?

Response: Thank you very much for your careful check of our manuscript. We have added error bars for OD for all figures.

  1. Line 204-205 describe the concentration of ethanol, methanol, 1-propanol, 1-butanol.

Response: Each alcohol was used at a final concentration of 0.1%. The concentrations have been described in Figure 3.

  1. Line 208-209, describe the concentration of the substrate added in the BM.

Response: As described above, the concentrations of all additives have been described in each figure.

Reviewer 2 Report

Dear Authors

great contribution on your continous work of oligothrophic strains.

some minor comments which I hope help:

- growth in inorganic basal medium indicates the use of CO2 as later mentioned; this can be clarified already in the abstract as it is known from your work for many years already.

- avoid duplication of phrases between title and keywords

- change gram-positive to "Gram-positive" as it is a proper name

- the suppl file comprises just 6 primers in a small table; this should be added to the methods section. I will not take too much space.

Author Response

To reviewer 2

Thank you very much for your useful comments concerning our manuscript. We have carefully revised the manuscript based on your suggestions.

Below, please find our responses to your specific comments.

  1. growth in inorganic basal medium indicates the use of CO2 as later mentioned; this can be clarified already in the abstract as it is known from your work for many years already.

Response: Thank you for your suggestion. We have added the description for the oligotrophic CO2 requirement of N9T-4 in the abstract (line 12).

  1. avoid duplication of phrases between title and keywords

Response: Based on your suggestion, we have considered keywords in our manuscript again (line 24).

  1. change gram-positive to "Gram-positive" as it is a proper name

Response: We have corrected the description as your suggestion (lines 161 and 166)

  1. the suppl file comprises just 6 primers in a small table; this should be added to the methods section. I will not take too much space.

Response: We have added the primer sequences in the Material and Methods section (line 96-109).